# Relevant Day/Night Temperatures Simulating Belgian Summer Conditions Reduce Japanese Encephalitis Virus Dissemination and Transmission in Belgian Field-Collected *Culex pipiens* Mosquitoes

**DOI:** 10.3390/v15030764

**Published:** 2023-03-16

**Authors:** Claudia Van den Eynde, Charlotte Sohier, Severine Matthijs, Nick De Regge

**Affiliations:** 1Exotic and Vector-Borne Diseases, Sciensano, Groeselenberg 99, 1180 Brussels, Belgium; 2Viral Re-Emerging Enzootic and Bee Diseases, Sciensano, Groeselenberg 99, 1180 Brussels, Belgium

**Keywords:** Japanese encephalitis virus, vector competence, field-caught mosquitoes, *Culex pipiens*

## Abstract

Japanese encephalitis virus (JEV), a zoonotic mosquito-borne Flavivirus, can be considered an emerging infectious disease. Therefore, vector competence studies with indigenous mosquitoes from regions where JEV is not yet endemic are of great importance. In our study, we compared the vector competence of *Culex pipiens* mosquitoes emerged from Belgian field-caught larvae under two different temperature conditions: a constant 25 °C and a 25/15 °C day/night temperature gradient representing typical summer temperatures in Belgium. Three- to seven-day-old F0-generation mosquitoes were fed on a JEV genotype 3 Nakayama strain spiked blood-meal and incubated for 14 days at the two aforementioned temperature conditions. Similar infection rates of 36.8% and 35.2% were found in both conditions. The observed dissemination rate in the gradient condition was, however, significantly lower compared to the constant temperature condition (8% versus 53.6%, respectively). JEV was detected by RT-qPCR in the saliva of 13.3% of dissemination positive mosquitoes in the 25 °C condition, and this transmission was confirmed by virus isolation in 1 out of 2 RT-qPCR positive samples. No JEV transmission to saliva was detected in the gradient condition. These results suggest that JEV transmission by *Culex pipiens* mosquitoes upon an accidental introduction in our region is unlikely under current climatic conditions. This could change in the future when temperatures increase due to climate change.

## 1. Introduction

Japanese encephalitis virus (JEV) belongs to the genus Flavivirus of the family Flaviviridae [1] and is a zoonotic, mosquito-borne virus that is maintained in a transmission cycle between mosquito vectors and vertebrate hosts, mainly Ardeid birds (herons and egrets) and pigs. These hosts produce high viremias [1] allowing mosquitoes to become infected when taking a blood meal. Humans, cattle and horses are considered dead-end hosts, as JEV infection results in insufficient viremia to infect naive mosquitoes when taking a blood meal. Nevertheless, infection of these hosts can lead to encephalitis with fever, tremors, convulsions, coma and death [2]. In humans, and mostly in children [3], 1% of infected individuals will develop encephalitis, with a mortality rate in this group of 20–30% [4]. JEV is the leading cause of viral encephalitis in many countries in Asia with 68,000 cases reported annually [5]. With the frequent occurrence of neurological sequelae, JEV has been said to cause more disability-adjusted life years than any other arbovirus. In addition, no specific treatment is available [6]. JEV is currently endemic in Australia (Torres Strait islands) and Southeast and East Asia, including the temperate zone of north-eastern China, Japan and Korea [7,8], making up nearly half of the human population lives in countries at risk [9]. In 2022, it spread to new regions in Australia [10]. JEV has already been detected outside its endemic areas, namely in an autochthonous case in Angola [11]. Additionally, JEV RNA was detected in a pool of *Culex pipiens* mosquitoes in Italy, although the presence of the virus has never been confirmed [12].

Current knowledge about vector competence and vector capacity of mosquitoes for JEV, as well as the limited knowledge about the underlying mechanisms affecting these parameters, was recently reviewed by our group [8]. Seventeen species have been reported to be able to transmit JEV in the laboratory and have additionally already been found positive in the field, namely *Aedes albopictus, Aedes vexans, Aedes vigilax, Anopheles tessellatus, Armigeres subalbatus, Culex annulirostris, Culex bitaeniorhynchus, Culex fuscocephala, Cullex gelidus, Culex pipiens, Culex pipiens pallens, Culex pseudovishnui, Culex quinquefasciatus, Culex sitiens, Culex tarsalis, Culex tritaeniorhynchus* and *Culex vishnui*, making them the currently known vectors for JEV. Of the aforementioned species, *Culex tritaeniorhynchus* and *Culex annulirostris* are the primary vectors in endemic areas. In addition, the following 10 species are potential vectors, as they have been proven competent in the laboratory, but without detection of JEV in field-collected species: *Aedes detritus, Aedes dorsalis, Aedes japonicus, Aedes kochi, Aedes nigromaculis, Aedes notoscriptus, Culiseta annulata, Culiseta incidens, Culiseta inornata,* and *Verrallina funerea.*

JEV is considered a potentially emerging infectious disease, and the likelihood of introductions increases with increasing globalization. One possible scenario for introduction is that imported infected mosquitoes lead to infection of susceptible animals in new areas. Another possibility is that viremic animals are imported. Subsequently, native mosquitoes may become infected by taking a blood meal from these infected animals and transmit JEV to new hosts [8]. Vector competence studies are therefore important to assess the risk that mosquito species occurring in non-endemic areas could transmit JEV in the event of an introduction. 

Different European mosquito species have been tested for their competence for JEV. *Aedes albopictus* and *Culex pipiens* were found competent in France [13], and *Aedes detritus, Culex pipiens* and *Culiseta annulata* in the UK [14,15,16]. Additionally, the detection of JEV RNA in Italy in 2010 occurred in *Culex pipiens* mosquitoes [12]. Moreover, *Culex pipiens* is already a known vector in several endemic areas like China [7] and Korea [17,18,19]. The aim of this study was to examine the vector competence of Belgian *Culex pipiens* mosquitoes, since this is the most abundant mosquito species in Belgium [20] and their host preference aligns with the JEV transmission cycle; in fact, biotype pipiens are strongly avian-seeking, which could associate them with viremic Ardeid birds. Biotype molestus would be more likely to feed on mammalian hosts [21], which might lead them to feed on a viremic pig. Consequently, *Culex pipiens* could contribute significantly to the spread of JEV if our native populations are competent. 

Furthermore, we studied vector competence at two temperature conditions, namely a constant 25 °C and a 25/15 °C day/night temperature gradient. This should provide more insight in the vector capacity of this species for JEV. Vector capacity, in addition to vector competence, takes into account additional factors such as environmental, behavioral, cellular and biochemical variables [22], making it more specific to the vector population at the prevailing climatic conditions in a given area at a given time. The choice of that specific temperature gradient was based on the fact that it closely matches the summer temperatures in Belgium, as in 2020 average summer maximum and minimum temperatures varied between 26.1 °C and 13.1 °C, respectively [23].

Studying the vector competence of field-collected *Culex pipiens* mosquitoes for JEV under different temperature conditions will lead to a reliable risk assessment of the role *Culex pipiens* could play in the spread of JEV upon an introduction in our region.

## 2. Materials and Methods

### 2.1. Mosquito Collection

Larvae of *Culex pipiens* were collected at several locations in Belgium (Ghent and its surrounding villages, Uccle and Zottegem). The field-collection was carried out during the vector season (May–October) of 2021 and 2022. Larvae were reared in plastic containers with water from the collection site supplemented with brewer’s yeast tablets (Holland & Barrett, Belgium) and fish food (Amazon, Belgium). Larvae were kept at 24 °C, 70% relative humidity with a 16 h:8 h light-dark cycle and when adults emerged, a 10% sucrose solution was provided. 

### 2.2. Virus Production and Titration

JEV genotype 3 Nakayama strain was used in our experiments. An eight passage of the virus was produced on 80% confluent Vero cells. Vero cells were kept at 37 °C in Dulbecco’s Modified Eagle Medium (DMEM) (ThermoFisher, Belgium) supplemented with 1% antibiotics (Gentamicin 50 mg/mL, ThermoFisher, Belgium; Antibiotic Antimycotic Solution 100×, Sigma-Aldrich, Belgium) and 10% fetal bovine serum (FBS) (Merck Life Science, Belgium). The supernatant was collected at 72 h post incubation and after three freeze/thaw cycles. A virus titration on Vero cells grown in a 96 well plate was performed to determine the titer of the virus stock, which was measured at 10^7.7^ TCID50/mL.

### 2.3. Oral Infection of Mosquitoes

The oral infection, the dissection and salivation were done according to standardized conditions as reviewed in [8]. Three- to seven-day-old F0 generation *Culex pipiens* mosquitoes were deprived of sucrose for 48 h. Subsequently, mosquitoes were transported to our BSL-3 facilities where all following steps were carried out according to the necessary regulations. Mosquitoes were allowed to feed on virus-spiked blood at a ratio of 1:2 (500 µL virus suspension to 1 mL blood) for 1 h using the Hemotek system (Hemotek Ltd., UK) which heated the spiked blood to 37 °C. Pig intestine (Butcher Shop Burms, Ghent) was used as a membrane. The infectious blood meal consisted of chicken blood (obtained from chickens housed at Sciensano or Van-O-Bel poultry slaughterhouse, Belgium), viral suspension (JEV genotype 3 Nakayama strain), and ATP (final concentration 5 μM) (Merck Life Science, Belgium). The viral titer in the blood meal was 10^6.39^ TCID50/mL at the start and remained >10^5.5^ TCID50/mL after 1 h of feeding. After blood feeding, mosquitoes were cold anaesthetized in a petri dish on ice, and blood-fed females were selected and transferred to a bugdorm mosquito cage. These blood-fed mosquitoes were kept either at a constant 25 °C or at a 25/15 °C day/night temperature gradient. At the gradient condition the temperature increased/decreased gradually over 4 h, 5 °C in the first hour and 5 °C in the next three hours. Additionally, light intensity was increased/decreased during this 4 hour-period. For both temperature conditions, a 70% relative humidity with a light/dark cycle of 16/8 h was implemented for 14 days. A 10% sucrose solution was provided during the 14 day-incubation period. 

### 2.4. Mosquito Salivation and Dissection

At 14 dpi (days post infection), JEV exposed mosquitoes were cold anaesthetized, legs and wings were removed and preserved in 500 μL DMEM containing 2% antibiotics (Antibiotic Antimycotic Solution 100×), to determine the dissemination rate of the virus. The bodies were attached to a glass slide using double-sided tape and modelling clay (Figure 1) and the proboscis was manually inserted into a 10 μL pipette tip filled with 5 μL FBS with 10% sugar. After 30 min, the content of the tip was transferred into an Eppendorf containing 45 μL DMEM with 2% antibiotics and 2% FBS to determine the transmission rate. A visual check was performed to control whether salivation had occurred. Mosquitoes showing an enlarged abdomen after the 30 min salivation period, due to the ingested FBS solution, were considered to have salivated. Subsequently, the head was separated from the body and added to the legs and wings to determine the dissemination rate. The body was kept separately in 500 μL DMEM with 2% antibiotics to determine the infection rate. All samples were stored at −80 °C until further analyses. 

### 2.5. JEV Detection 

#### 2.5.1. RT-qPCR Analysis

Stainless steel beads were added to the mosquito bodies and to the head-wings-legs in DMEM with antibiotics, and homogenization was done using a Tissuelyser (Tissuelyser II, Qiagen, Belgium). RNA was extracted using the QIAamp viral RNA mini kit (Qiagen, Belgium) according to the manufacturer’s protocol. RNA was used directly for RT-qPCR analysis or stored at −80 °C until use. The RT-qPCR mixture contained AgPath-ID™ One-Step RT-PCR Reagents (ThermoFisher, Belgium), 800 nM of each of the JEV NS2A primers (forward: 5′-AGCTGGGCCTTCTGGT-3′ and reverse: 5′-CCCAAGCATCAGCACAAG-3′) and 400 nM of the probe (Fam-CTTCGCAAGAGGTGGACGGCCA-Tamra) (Integrated DNA Technologies, Belgium) [24], and 5 μL RNA sample. Samples were run on a LightCycler480 according to the following temperature program: 45 °C for 10 min and 95 °C for 10 min; followed by 45 cycles at 95 °C for 15 s, and 60 °C for 45 s. CT-values ≤ 40 and with curves that showed an exponential amplification were considered positive. 

#### 2.5.2. Virus Isolation

Samples found positive by RT-qPCR were tested in virus isolation. This involved transferring 50 μL of the homogenate or saliva in DMEM onto a well in a 96-well plate containing 80% confluent Vero cells. After inoculation, the cells were incubated at 37 °C for 4 h before subsequently adding 100 μL DMEM with 2% antibiotics and FBS (final concentration 10%) and incubating them for 7 days at 37 °C. On day 7, a second passage on Vero cells was performed by transferring 50 μL of the supernatant to a new 96-well and repeating the protocol described above. After another 7-day incubation period, the plates were fixed by adding methanol (−20 °C) to the cells for 20 min; after removal of the methanol, the plates were placed at −20 °C for at least 24 h to evaporate any remaining methanol. Thereafter, virus detection was achieved by fluorescent staining with a primary NS1 antibody (Viral Japanese Encephalitis virus NS1 Antibody, Bio-Techne, UK, Catalog # MAB100061) and secondary Alexa Fluor^®^ 488 anti-mouse IgG2b Antibody (BioLegend, The Netherlands, Catalog # 406718).

### 2.6. Biotype Identification by qPCR

To identify the *Culex pipiens* biotype, and to distinguish them from the morphologically identical species *Culex torrentium*, the extract from the body homogenates was used in an additional qPCR with biotype-specific probes. The protocol for the qPCR assay was adopted from Vogels, 2017 [25]. Briefly, primers targeting DNA sequences of microsatellite CQ11 were used, i.e., Cx_pip_F (5′-GCGGCCAAATATTGAGACTTTC-3′) and Cx_pip_R (5′-ACTCGTCCTCAAACATCCAGACATA-3′), as universal *Culex pipiens* primers. Probe Cpp_mol_P (5’-FAM-TGAACCCTCCAGTAAGGTA-3’) was used for the identification of biotype molestus and probes Cpp_pip_P1 (5′-FAM-CACACAAAYCTTCACCGAA-3′) and Cpp_pip_P2 (5′-FAM-ACACAAACCTTCATCGAA-3′) (Integrated DNA Technologies, Belgium) for identification of biotype pipiens. qPCR testing was performed for each biotype in separate reactions and the mix contained SsoAdvanced universal Probes supermix (2x) (Bio-Rad, Belgium), 300 nM of each of the primers (Cx_pip_F and Cx_pip_R), and 200 nM of the Cpp_pip1 probe or 100 nM of the Cpp_pip_P2 or Cpp_mol_P probe. Nuclease-free water and 5 μL of RNA were added to arrive at a final volume of 20 μL. Samples were run on a LightCycler480 according to the following temperature program: 95 °C for 3 min, followed by 45 cycles of 95 °C for 15 sec and 60 °C for 45 s. CT-values ≤ 40 and with curves that showed an exponential increase were considered positive. 

### 2.7. Statistical Analysis

Fisher’s exact tests were used to determine whether infection, dissemination and transmission rates differed significantly between the two temperature conditions. Unpaired t-tests were used to determine whether significant differences in the CT-values occurred between the two temperature conditions and between the CT-values in infection from dissemination positive and negative mosquitoes in the 25 °C condition. Linear regression was implemented to see whether a correlation existed between the CT-values in infection and dissemination at the 25 °C condition. Statistical analyses were done using GraphPad Prism 9. *p*-values < 0.05 were considered to be significant.

## 3. Results

### 3.1. Infection, Dissemination and Transmission Rates

#### 3.1.1. Incubation at a Constant 25 °C Temperature

Out of approximately 1000 *Culex pipiens* mosquitoes, which emerged from field-collected larvae, that were allowed to feed on a blood meal containing JEV, 95 mosquitoes fed (9.5%). These were subsequently incubated for 14 days at 25 °C, 70% relative humidity and an 16/8 h light/dark cycle. Seventy-six (80%) *Culex pipiens* survived the 14-days incubation period (Table 1). The ratio of pipiens to molestus biotype at this condition was 12/63, and remarkably one mosquito tested positive for both biotypes and can therefore be interpreted as a hybrid species. Of the 76 surviving mosquitoes, 28 mosquito bodies were RT-qPCR positive for JEV, including one molestus and the one hybrid mosquito, leading to an infection rate of 36.8% (Table 1; Figure 2). Of these 28 infected mosquitoes, 15 were positive for JEV in wings/legs/heads, all belonging to the pipiens biotype, leading to a dissemination rate of 53.6%. Of the dissemination-positive mosquitoes, two were found positive by RT-qPCR in saliva which thus gives us a transmission rate of 13.3%. However, only six of the mosquitoes with a disseminated infection showed visible evidence of salivation. For that reason, if we only consider these mosquitoes with visual proof of salivation, a transmission rate of 33.3% was found. The overall transmission efficiency (i.e., positive saliva samples upon total mosquitoes blood-fed) was 2.6% for *Culex pipiens*, which may be considered the minimal transmission rate.

#### 3.1.2. Incubation at a 25/15 °C Day/Night Temperature Gradient

To assess the vector competence at a 25/15 °C day/night temperature gradient, approximately 600 *Culex pipiens* were exposed to a blood meal containing JEV, of which 111 mosquitoes (18.5%) fed. Seventy-one (64%) *Culex pipiens* survived the 14-day incubation period. The ratio of pipiens to molestus biotype at this condition was 6/65. The infection rate determined by RT-qPCR was 35.2% (25/71), including one molestus biotype mosquito, similar as found in the 25 °C constant condition. The dissemination rate was however significantly lower (Fisher’s exact test, *p*-value 0.0004) than in the constant 25 °C condition, namely only 8% (2/25). None of these two dissemination-positive mosquitoes tested positive for JEV in their saliva, leading to a transmission rate and transmission efficiency of 0% (Table 1; Figure 2).

### 3.2. Virus Isolation

JEV RT-qPCR positive samples were subject to virus isolation. The majority (77.8%) was confirmed positive (Table 2). Importantly, one RT-qPCR positive saliva sample from a mosquito incubated at the constant 25 °C condition was positive in virus isolation (Figure 3), thereby confirming the presence of infectious virus in the saliva. 

### 3.3. Viral Loads Found in Infection, Dissemination and Transmission

Viral loads in samples tested for infection (based on obtained CT-values) varied greatly between individual mosquitoes in both temperature conditions and no significant difference was found between the mean CT’s of both conditions (27.93 and 29.27; *p*-value 0.3129) (Figure 4A). In the constant 25 °C condition, also no significant difference was found between the mean value for infection in mosquitoes with (mean CT 27.93) and without (mean CT 28.61) a disseminated infection (*p*-value 0.5887), indicating that the viral load in infection does not predict whether JEV will disseminate or not.

When looking at the viral load in samples for dissemination (Figure 4B), also an important variation in CT-values between mosquitoes was observed. Interestingly, the two mosquitoes that were positive for JEV in saliva had the lowest CT-values in dissemination, indicating a viral load-dependent passage of the salivary gland escape barrier.

Furthermore, a linear regression analyses was performed between the CT-values in infection and dissemination of the individual mosquitoes at the constant 25 °C temperature condition (Figure 5). The slope of the regression (0.2445) was not significantly different from zero (*p*-value: 0.1956), indicating there is no correlation between viral load in the midgut and the rest of the body (head, wings and legs) measured at day 14 post feeding.

## 4. Discussion

Globalization increases the chance that pathogens get introduced in new regions. JEV is considered an emerging disease, and therefore we studied whether Belgian *Culex pipiens* would be able to transmit JEV upon introduction. Given that this species was already found competent in other regions [7,17,18,19] and that it is the most abundant species in Belgium [20,26] it could contribute significantly to the spread of JEV. We decided to study this with adult mosquitoes emerging from field-caught larvae since these reflect natural populations, possessing important genetic diversity and larval habitat specific microbiota. This makes the obtained results highly relevant. Working with field-caught mosquitoes, however, poses additional challenges to perform vector competence studies: they are not available all year, they show a lower blood feeding rate, and they have a noticeably higher mortality rate during incubation than colonized mosquitoes, making it more difficult and time consuming to obtain sufficient infected mosquitoes to produce relevant results. Colonized mosquitoes are already adapted to the laboratory environment and to artificial drinking systems, which leads to higher survival and feeding rates [27,28].

In our study we showed that *Culex pipiens* is able to transmit JEV at a constant 25 °C condition, this with an overall transmission efficiency of 2.6%. When our obtained infection, dissemination and transmission rates were compared with these of vector competence studies conducted with similar titers of genotype 3 strains in our neighboring countries, we see that our rates are remarkably lower. For example, in the UK [15], a colonized line of *Culex pipiens* biotype pipiens mosquitoes (F > 100 generations) reached a 90% (18 out of 20) infection rate, a 77.8% (14 out of 18) dissemination rate and even a 100% (14 out of 14) transmission rate at a 25 °C condition. Another vector competence study with colonized *Culex pipiens* mosquitoes conducted in France by de Wispelaere et al. [13] also reported higher rates at a constant 26 °C temperature condition, namely 80%, 70% and 42% infection, dissemination and transmission rates. We hypothesize that these clear differences between our field-collected mosquitoes and colonized mosquitoes are due to microbiome changes and/or a lower genetic diversity. These differences between field-collected and colonized mosquitoes have already been reported for other mosquito species. The bacterial diversity in field-collected *Aedes albopictus* mosquitoes was found to be much higher compared to colonized lines [29]. Additionally, it was demonstrated that colonized *Aedes aegypti* mosquitoes are less genetically variable than their field counterparts [30]. 

Since a constant 25 °C temperature is not representative for current summer conditions in Belgium or other northern European countries, we also studied the effect of a 25/15 °C day/night temperature gradient on the vector competence of *Culex pipiens* for JEV. Importantly, in these conditions *Culex pipiens* was not able to transmit JEV after an incubation period of 14 days. The major difference between both temperature conditions was found for the dissemination rate, implying that the virus was severely hampered to cross the midgut in the gradient condition, indicating that crossing of the midgut escape barrier is temperature-dependent. A negative impact of low temperatures on vector competence for JEV was reported before, with lower infection, dissemination and transmission at 20 °C compared to 25 °C [15]. The negative effect of a lower temperature on midgut escape has also been demonstrated *in Culex pipiens* for West Nile virus [31,32]. The mechanistic explanation for this observation however remains elusive. It might be due to a shift in the microbiome or to changes in the metabolism of mosquitoes who are ectothermic [33]. More hypotheses have recently been reviewed by Samuel et al. [34], highlighting the effect of temperature on the balance between virus replication and the immune response (RNAi) of the vector. In this context, it is remarkable that Chapman et al. [14] reported an infection rate of a 100% (18 out of 18) and a transmission rate of 72.2% (13 out of 18) in field-collected *Culex pipiens* mosquitoes (from Neston, UK) incubated at a constant 18 °C. These results could be related to the relatively low number of mosquitoes tested or to the fact that a JEV genotype 2 strain was used, but further investigations seem warranted. Moreover, it would be interesting to determine at which minimum temperature of a gradient, dissemination ceases to be significantly different lower than at a constant 25 °C temperature condition, from which temperature the midgut escape barrier is breached more easily, or also to know what the lowest temperature is at which transmission is still possible. As a last point, all vector competence parameters in this study were determined after an incubation time of 14 days. It would be interesting to evaluate whether these would remain similar after a longer incubation period, e.g., 21 days, and study whether the dissemination is only delayed in the gradient condition rather than inhibited.

When considering the vector competence of the *Culex pipiens* biotypes separately, one molestus biotype was found positive in infection at each temperature condition. Moreover, at the constant temperature, the single hybrid species mosquito also tested positive in infection. JEV did not disseminate in any of these three mosquitoes, making them not competent. However, it is not possible to draw a conclusion for the entire molestus or hybrid biotype given the low numbers tested, namely 18 molestus biotype (12 at the constant, 6 at the gradient condition) and one hybrid species.

RT-qPCR positive samples were tested in isolation and the majority (77.8%) was confirmed positive, showing the presence of infectious virus. Isolation is a less sensitive technique than RT-qPCR [35,36], which may explain the samples that were not confirmed. Most importantly JEV was isolated from a saliva sample of one of the two positive RT-qPCR samples at the constant 25 °C condition. This provides further evidence that JEV could be transmitted by infected *Culex pipiens* mosquitoes when these take a second blood meal.

When looking at viral loads, we observed no significant differences in CT-values upon infection under both conditions. This was somewhat unexpected as Folly et al. [15] reported significantly higher loads in infection at 25 °C compared to 20 °C conditions. This might be due to the fact that we used a temperature gradient whereby the mosquitoes were a considerable number of hours per day at 25 °C, allowing efficient virus replication. Subsequently, we compared the CT-values of infection of mosquitoes with proven dissemination with those in which JEV did not disseminate, finding no significant difference, suggesting that crossing of the midgut barrier is independent of viral load in the midgut. This has been observed before and was recently thoroughly described in the review by Carpenter et al., 2023 [37]. They state that it is unclear whether replication in the midgut is in fact necessary for a successful dissemination. The appearance of viruses in the hemolymph was namely observed at times before the virus had sufficient time to replicate [38,39]. Other studies however indicated that certain viral loads must be reached in the midgut before Western equine encephalitis virus and Zika virus can disseminate in *Culex* and *Aedes albopictus* mosquitoes, respectively [40,41].

Besides the midgut barrier, salivary gland barriers also play an important role in vector competence. Although our data are limited, the two transmission positive mosquitoes at the 25 °C condition were the ones with the lowest CT-values in dissemination, suggesting that crossing of the salivary gland barriers is only possible when the viral load is sufficiently high. This phenomenon was also pointed out in a study of Sanchez-Vargas et al. [42] using *Aedes aegypti* infected with Chikungunya, Dengue or Zika virus. They state that when the amount of virus is low, genetic factors in a mosquito influence the viral titer in the salivary glands. However, once the viral titer is high enough in the salivary glands, the genetic contribution of the mosquito no longer has any effect. Therefore, higher viral loads in the salivary gland cells (indicated in our study by a low CT-value in dissemination) may lower the mosquitoes’ immunity, leading to the crossing of the salivary gland escape barrier, and thus the presence of virus in saliva and the potential for transmission.

Based on the results of our vector competence study, JEV transmission seems unlikely when relevant day/night temperature conditions for a Belgian summer are simulated. Although vector competence studies cannot be extrapolated as such to field conditions, as many other aspects making up the final vectorial capacity come into play, these results suggest that the actual risk that Belgian *Culex pipiens* mosquitoes could transmit JEV upon an introduction is low. This could change if global warming increases temperatures to a more constant 25 °C. However, we should note that even in these conditions, the obtained transmission efficiency in our study was low (2.6%). This low rate might however be an underestimation as not all mosquitoes salivate within the 30-min period. In fact, if we only look at the dissemination positive mosquitoes which visibly salivated, the transmission rate increases from 13.3 to 33.3%. In a study by Heitmann et al. [43], it was even suggested to use mosquito leg screens to evaluate transmission potential, as they stated that forced salivary collection methods tend to underestimate the transmission rate. Such leg screens, however, take no account whatsoever of the salivary barriers, making it less relevant and might even overestimate transmission. However, even with a low overall transmission rate, *Culex pipiens* could be a high-capacity vector for JEV. It is namely the most common mosquito species in Belgium that can be found in high abundance in forested, rural and urban areas, implying that this competent vector can reside in almost any habitat. Furthermore, their host preference aligns with the replication cycle of JEV, as they can feed on pigs and Ardeid birds and thus support its transmission between susceptible hosts.

## 5. Conclusions

Vector competence studies with field-collected mosquitoes are highly relevant but challenging due to low feeding rates, time-consuming field collection and sub-optimal surviving rates of adults after infection. In our study we have proven that Belgian *Culex pipiens* are competent vectors for JEV when kept at a constant 25 °C. When this temperature was, however, changed to a 25/15 °C day/night temperature gradient, a significantly lower dissemination was found and no transmission. This suggests that under current conditions, the risk that *Culex pipiens* mosquitoes could transmit JEV upon an introduction in Belgium seems low, but this could evolve when climate change would lead to higher temperatures during summer or extreme weather events involving high temperatures for extended periods of time. 

## Figures and Tables

**Figure 1 viruses-15-00764-f001:**
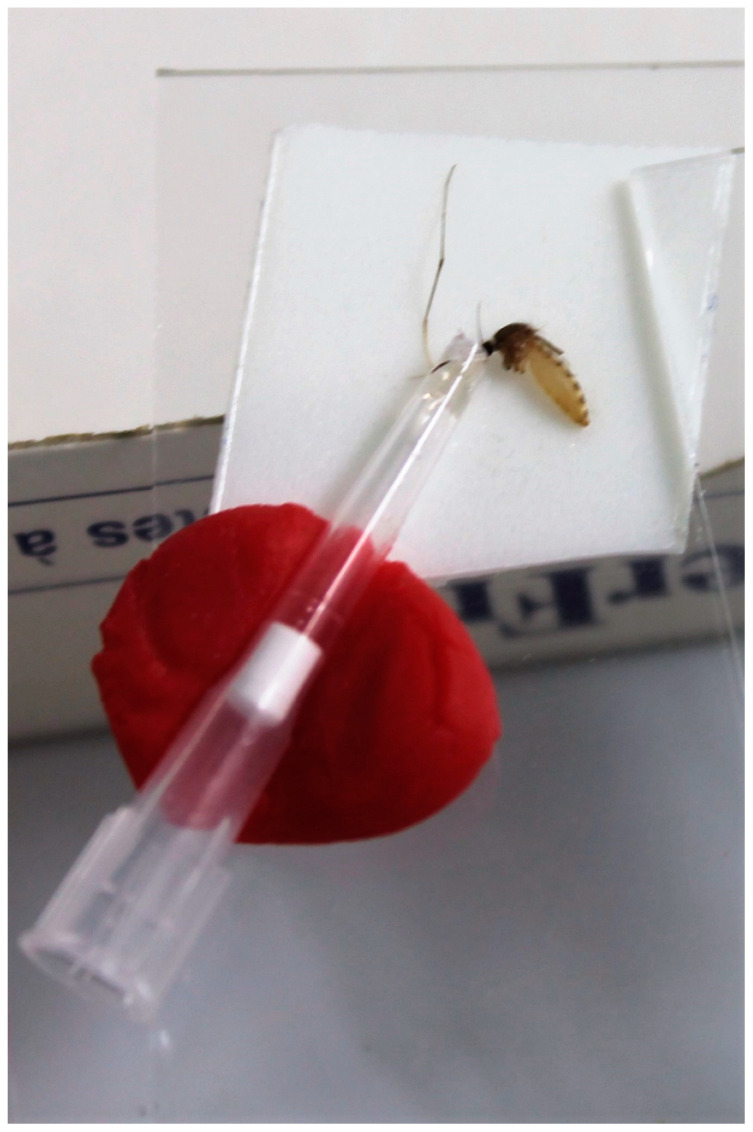
Mosquito salivation set-up. Mosquito bodies (without legs and wings) were attached to double-side tape and their proboscis was manually inserted in a pipette tip containing FBS + sugar.

**Figure 2 viruses-15-00764-f002:**
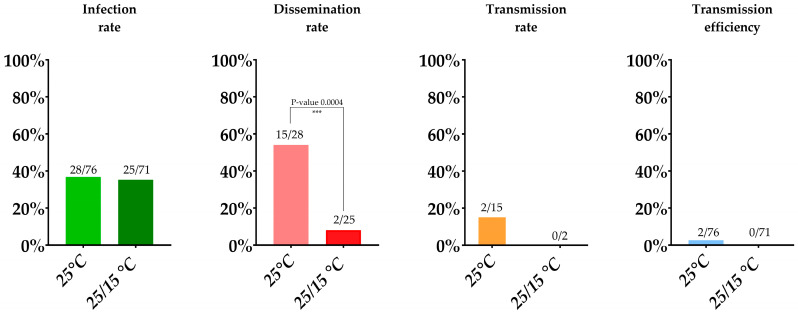
Infection rate, dissemination rate, transmission rate, and transmission efficiency at the two temperature conditions. Absolute numbers are displayed above bars. Fisher’s exact tests were performed to check differences between rates at both temperature conditions.

**Figure 3 viruses-15-00764-f003:**
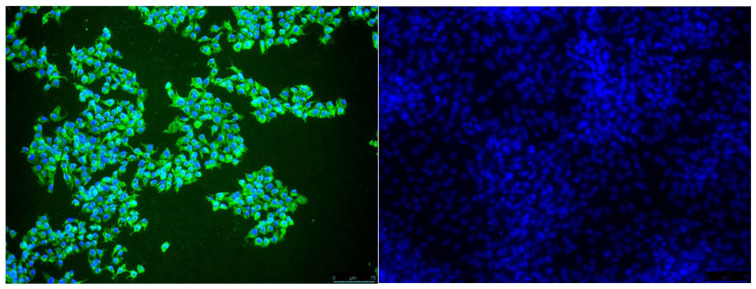
Japanese encephalitis virus isolation from mosquito saliva. JEV was isolated from the saliva of a *Culex pipiens* mosquito collected at 14 dpi from the constant 25 °C temperature condition (left panel). Saliva of a non-blood fed mosquito was included as a negative control (right panel). The second passage on Vero cells was stained for JEV with a primary anti-JEV NS1 antibody and secondary Alexa Fluor^®^ 488 anti-mouse IgG2b Antibody (green). Cell nuclei were stained with Hoechst (blue).

**Figure 4 viruses-15-00764-f004:**
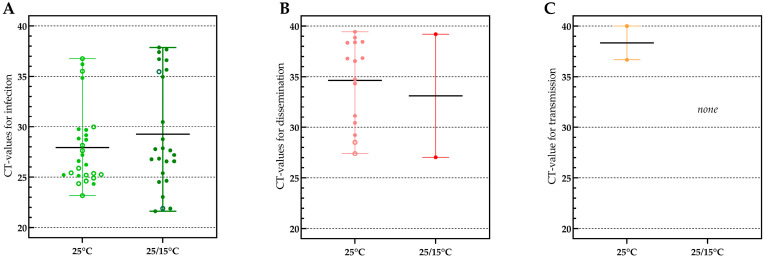
Viral loads (CT-values) found in samples of *Culex pipiens* mosquitoes found positive for infection (**A**), dissemination (**B**) and transmission (**C**). Mean CT’s are indicated by horizontal black lines. Open circles in panel A and B indicate individual mosquitoes that tested positive in dissemination and transmission, respectively. Filled dots in panel A and B indicate individual mosquitoes that tested negative in dissemination and transmission, respectively.

**Figure 5 viruses-15-00764-f005:**
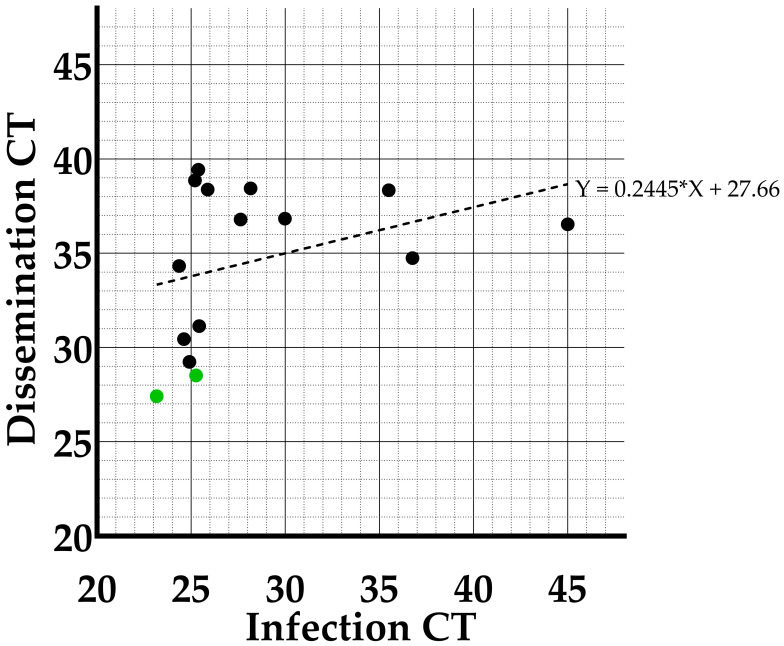
Graph showing the correlation between CT-values for infection and dissemination in *Culex pipiens* at the constant 25 °C temperature condition implemented during incubation. Green dots indicate individual mosquitoes which tested positive in transmission.

**Table 1 viruses-15-00764-t001:** Vector competence of *Culex pipiens* for JEV at two temperature conditions. Results are based on RT-qPCR analyses. Rates are mentioned in percentages and absolute numbers are mentioned between brackets.

	14 Day Survival Rate	Infection Rate	Dissemination Rate	Transmission Rate	Transmission Efficiency
25 °C	80% (76/95)	36.8% (28/76)	53.6% (15/28)	13.3% (2/15)	2.6% (2/76)
25/15 °C	64% (71/111)	35.2% (25/71)	8% (2/25)	0% (0/2)	0% (0/71)

**Table 2 viruses-15-00764-t002:** Virus isolation of RT-qPCR-positive samples for infection, dissemination and transmission. All ratios are given in percentages with the respective absolute numbers between brackets.

	Infection RT-qPCR Positives Confirmed by Isolation	Dissemination RT-qPCR Positives Confirmed by Isolation	Transmission RT-qPCR Positives Confirmed by Isolation
25 °C	71.4% (20/28)	86.7% (13/15)	50% (1/2)
25/15 °C	92% (23/25)	100% (2/2)	*No RT-qPCR positive samples*

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
