# Peer review of "Relevant Day/Night Temperatures Simulating Belgian Summer Conditions Reduce Japanese Encephalitis Virus Dissemination and Transmission in Belgian Field-Collected Culex pipiens Mosquitoes"

_viruses, 2023, doi:10.3390/v15030764_

Round 1
Reviewer 1 Report
This is a straightforward and solid manuscript. It is well-written and clear in scope and presentation. Results are important and only minor additions (see below) are suggested.
Results:
- As only ~10% of the mosquitoes fed, the main conclusion (extrapolation) of the manuscript, that transmission is unlikely to occur under the current climatic condition, can be compromised. The 10% that fed may not represent the majority of the population (90%), and therefore its vector competence. Authors should discuss and highlight the important differences between the conditions to sustain infection, dissemination and transmission occurring in the laboratory and the wild. For example, humidity was kept constant, even under the temperature variation setting. Also, I suggest increasing ATP (pH 7,4) as a phagostimulant up to 1mM to enhance feeding success.
- Another problem with the reduced numbers of fed mosquitoes is the even smaller number of mosquitos available for the transmission evaluation (Table 1 - 17 total).
- The fact that mortality is higher for the group under different temperature regimens (25/15) also reduces the extrapolation potential of the finding. It’s possible to think of a scenario where the group 25/15 is experiencing stress, compared to control, and this is reducing its vector competence. This should be considered and discussed.
- lines 201 “The ratio of pipiens to molestus biotype at this condition was 12/63, and remarkably one mosquito tested positive for both biotypes and can therefore be interpreted as a hybrid species”. Another interpretation is that the primers are not that specific.
- lines 211 “The overall transmission efficiency (i.e., positive saliva samples upon total mosquitoes blood-fed) was 2.6% for Culex pipiens, which may be considered the minimal transmission rate.” This low transmission rate occurred probably because day 14 was picked. It might be too early for a flavivirus. This should be discussed.
Author Response
We thank the reviewer for the constructive criticism. We have addressed all comments and made modifications to the manuscript as documented below and with track changes in the revised manuscript. If there are any suggestions for the revised version, please do not hesitate to contact us. We appreciate the thorough analysis and hope that this improved version will be published later in Viruses.
- As only ~10% of the mosquitoes fed, the main conclusion (extrapolation) of the manuscript, that transmission is unlikely to occur under the current climatic condition, can be compromised. The 10% that fed may not represent the majority of the population (90%), and therefore its vector competence. Authors should discuss and highlight the important differences between the conditions to sustain infection, dissemination and transmission occurring in the laboratory and the wild. For example, humidity was kept constant, even under the temperature variation setting. Also, I suggest increasing ATP (pH 7,4) as a phagostimulant up to 1mM to enhance feeding success.
As vector competence is a factor determined in the laboratory, direct extrapolation to the field (vectorial capacity) is not possible, nor was it the purpose of our study. We did however demonstrate that a different temperature affected the vector competence of Culex pipiens mosquitoes in laboratory conditions, which we believe will also affect vector competence in the field. To actually extrapolate to field conditions, vectorial capacity must be assessed (taking into account additional factors such as environmental, behavioural, cellular and biochemical variables).
We deleted the following sentence in the discussion (lines 294-295): and extrapolatable to actual field conditions; and added the following sentences (line 391-395): Based on the results of our vector competence study, JEV transmission seems unlikely when relevant day/night temperature conditions for a Belgian summer are simulated. Although vector competence studies cannot be extrapolated as such to field conditions, as many other aspects making up the final vectorial capacity come into play, these results suggest that the actual risk that Belgian Culex pipiens mosquitoes could transmit JEV upon an introduction is low.
For further experiments, it would indeed be interesting to have the humidity change with temperature, as the relative humidity will be highest when the temperature is coolest, and lowest when the air temperature is highest. We also thank the reviewer for the suggestion of increasing ATP and will implement this in further experiments.
- Another problem with the reduced numbers of fed mosquitoes is the even smaller number of mosquitos available for the transmission evaluation (Table 1 - 17 total).
In our study, we obtained about 70 blood-fed mosquitoes per condition, a number higher than many other vector competence studies. The fact that not all of these mosquitoes were successfully infected and an additional proportion were not positive for dissemination, leading to a small number of mosquitoes available for transmission, is explained by the lower competence of our field-caught mosquitoes. Namely, the denominator of dissemination (28 and 25) and transmission (15 and 2) is the number of mosquitoes that were positive for infection and dissemination, respectively. Therefore, we added transmission efficiency, since it represents the number of mosquitoes with JEV in saliva relative to the number of mosquitoes that fed, and thus indicates the likelihood that a mosquito can transmit JEV after ingesting an infected blood meal.
- The fact that mortality is higher for the group under different temperature regimens (25/15) also reduces the extrapolation potential of the finding. It’s possible to think of a scenario where the group 25/15 is experiencing stress, compared to control, and this is reducing its vector competence. This should be considered and discussed.
Even though mortality was higher at the temperature gradient condition, we still had 71 mosquitoes after 14 days for further analyses, which should be representative as we expect to obtain similar vector competence rates if more mosquitoes would have survived.
- lines 201 “The ratio of pipiens to molestus biotype at this condition was 12/63, and remarkably one mosquito tested positive for both biotypes and can therefore be interpreted as a hybrid species”. Another interpretation is that the primers are not that specific.
We thank the reviewer for their suggested interpretation. However, the primers were taken from the thesis of Chantal B.F. Vogels (The role of Culex pipiens mosquitoes in transmission of West Nile virus in Europe), who optimized the RT-PCR protocol from Rudolf et al. 2013. Additionally, we considered only the samples with an exponential curve as positive. Moreover, considering that there are several reports of hybrid species in the literature, the hybrid species in our study is definitely not unlikely.
- Amraoui, F., Tijane, M., Sarih, M. et al. Molecular evidence of Culex pipiens form molestus and hybrids pipiens/molestus in Morocco, North Africa. Parasites Vectors 5, 83 (2012). https://doi.org/10.1186/1756-3305-5-83
- Rudolf M, Czajka C, Börstler J, et al. First nationwide surveillance of Culex pipiens complex and Culex torrentium mosquitoes demonstrated the presence of Culex pipiens biotype pipiens/molestus hybrids in Germany. PLoS One. 2013;8(9):e71832. Published 2013 Sep 11. doi:10.1371/journal.pone.0071832
- Osório HC, Zé-Zé L, Amaro F, Nunes A, Alves MJ. Sympatric occurrence of Culex pipiens (Diptera, Culicidae) biotypes pipiens, molestus and their hybrids in Portugal, Western Europe: feeding patterns and habitat determinants. Med Vet Entomol. 2014;28(1):103-109. doi:10.1111/mve.12020
- lines 211 “The overall transmission efficiency (i.e., positive saliva samples upon total mosquitoes blood-fed) was 2.6% for Culex pipiens, which may be considered the minimal transmission rate.” This low transmission rate occurred probably because day 14 was picked. It might be too early for a flavivirus. This should be discussed.
Implemented, the following was added to the discussion (line 345-349): As a last point, all vector competence parameters in this study were determined after an incubation time of 14 days. It would be interesting to evaluate whether these would remain similar after a longer incubation period, e.g. 21 days, and study whether the dissemination is only delayed in the gradient condition rather than inhibited.
Reviewer 2 Report
Review: viruses-2269594
This manuscript by Van den Eynde et al described compared the vector competence of Culex pipiens mosquitoes emerged from Belgian field-caught larvae under two different temperature conditions. While the subject is appropriate for the journal, publication of this manuscript in its present form is not recommended, because it contains a number of unexplained observations.
To be considered further for publication the manuscript will need to be more organized (including adequate references) and elaborative in support of the claims made in the paper. Some specific points of concern are noted below:
1) There have, of course, been many other efforts to describe the structures of the WT and mutant species of JE viruses. These have largely been overlooked in the current work. There have been more papers on this virus and perhaps the bibliography or citation list could use a few more.
2) Brief statistical analysis is already included in this paper. However, a brief mention on Japanese encephalitis virus structure and perspectives on new treatments will improve the quality of this paper. Few examples are:
https://doi.org/10.1038/nrneurol.2018.30
https://doi.org/10.1007/s12026-020-09130-y
https://doi.org/10.3201/eid1501.080311
3) The discussion section seems scattered. It needs to be concise and structured.
Author Response
We thank the reviewer for the constructive criticism. We have addressed all comments and made modifications to the manuscript as documented below and with track changes in the revised manuscript. If there are any suggestions for the revised version, please do not hesitate to contact us. We appreciate the thorough analysis and hope that this improved version will be published later in Viruses.
1) There have, of course, been many other efforts to describe the structures of the WT and mutant species of JE viruses. These have largely been overlooked in the current work. There have been more papers on this virus and perhaps the bibliography or citation list could use a few more.
Although the reviewer's comment is certainly an interesting note, we believe that such information about the wild type and mutant types of JEV is beyond the scope of our article. Given that our experiments were conducted with only one strain (Nakayama) of genotype three, more information on other genotypes and variants/mutants of JEV would be more appropriate in a review on the virus itself rather than in an article on vector competence.
2) Brief statistical analysis is already included in this paper. However, a brief mention on Japanese encephalitis virus structure and perspectives on new treatments will improve the quality of this paper. Few examples are:
https://doi.org/10.1038/nrneurol.2018.30
https://doi.org/10.1007/s12026-020-09130-y
https://doi.org/10.3201/eid1501.080311
Implemented. The following was added to the introduction, using the suggested articles (line 39-41): and an estimated 3 billion people at risk. With the frequent occurrence of neurological sequelae, JEV has been said to cause more disability-adjusted life years than any other arbovirus. In addition, no specific treatment is available [6].
And lines 43-44: making that nearly half of the human population lives in countries at risk [9].
3) The discussion section seems scattered. It needs to be concise and structured.
Comments on the discussion have already been made by the other reviewers, we hereby refer to the revised manuscript. Should any unconcise or less structured sections remain in the present version, we would appreciate to receive further suggestions.
Reviewer 3 Report
The manuscript "Relevant day/night temperature conditions reduce Japanese encephalitis virus dissemination and transmission in Belgian field-collected Culex pipiens mosquitoes" reports a study comparing mosquitoes blood fed with JEV held at a constant temperature (25oC) and a second group exposed to varying temperatures (15/25oC) to reflect diurnal variation experienced in Belgium summers. The authors find evidence for infection, dissemination and transmission in mosquitoes held under constant conditions, whereas the varying group were infected, considerably less dissemination and no transmission at 14 days. They conclude that transmission of JEV in Belgium, if it were to be introduced, is low. Overall the experimental procedures are sound and the results support the final conclusions although there is nothing to suggest that experiments were repeated to confirm the findings. The main issue with this article is the tone of some of the statements within in it, particularly in the Discussion. The following comments are for the authors to consider.
1. In the title, the authors need to explain the term "relevant" - to whom or what?
2. The authors state that JEV has been detected in Europe. They are right that a paper has been published claiming this. However, the original article by Ravanini et al., states that JEV RNA (167bp) had been detected and that "The authors are aware that these findings are preliminary, and confirmation of the results is necessary". No such confirmation has ever been presented and yet this is now stated as fact. I would recommend the authors highlight the uncertainty on this claim.
3. In section 2.3, a minor observation but presumably only females were used in the study.
4. In the results, the authors state that the "ratio of pipiens to molestus biotype is 12/63", and again below. Is this the correct ratio?
5. Figure three requires an uninfected control for comparison to appreciate the staining pattern.
6. At the end of the 1st paragraph of the Discussion, suggest "feeding" rather than "drinking".
7. The authors state that they hypothesize that these clear differences between field-collected and colonized mosquitoes are due to microbiome changes and/or a lower genetic diversity. What exactly is this hypothesis based on? They then go on to state that "This has been proven for other mosquito-arbovirus combinations [27-30]." The references to the influence of the microbiome are reviews and a book chapter. These base these assumptions on experiments crudely treating mosquitoes with antibiotics. The authors should refer to original research that demonstrates that colonisation influences that mosquito microbiome (demonstrated rather than implied) to a degree that changes vector competence.
8. In the following paragraph there are numerous references to midgut escape barrier. What evidence do the authors have for this influencing virus replication in their model? Reduced temperature may have influenced virus replication although the implication of Figure 2 suggests that this is not the case. The lower temperature regime may have delayed virus egress rather than prevented it, but only one time point was used. Overall the findings of this study seem similar to Folly et al., 2021.
9. In the conclusions, the data "suggests" rather than "makes" that under current conditions. Also the authors should consider extreme weather events where temperatures are elevated for extended periods. This appears to be a consequence of climate change. Finally, another flavivirus, has been transmitted successfully in Belgium, can the authors comment?
Author Response
We thank the reviewer for the constructive criticism. We have addressed all comments and made modifications to the manuscript as documented below and with track changes in the revised manuscript. If there are any suggestions for the revised version, please do not hesitate to contact us. We appreciate the thorough analysis and hope that this improved version will be published later in Viruses.
- In the title, the authors need to explain the term "relevant" - to whom or what?
Implemented, the title was changed to: Relevant day/night temperatures simulating Belgian summer conditions reduce Japanese encephalitis virus dissemination and transmission in Belgian field-collected Culex pipiens mosquitoes.
- The authors state that JEV has been detected in Europe. They are right that a paper has been published claiming this. However, the original article by Ravanini et al., states that JEV RNA (167bp) had been detected and that "The authors are aware that these findings are preliminary, and confirmation of the results is necessary". No such confirmation has ever been presented and yet this is now stated as fact. I would recommend the authors highlight the uncertainty on this claim.
We thank the reviewer for this recommendation and the following was added to the introduction (line 47-48): Additionally, JEV RNA was detected in a pool of Culex pipiens mosquitoes in Italy, although the presence of the virus has never been confirmed.
- In section 2.3, a minor observation but presumably only females were used in the study.
To avoid unnecessarily subjecting the mosquitoes to a prolonged cold shock needed to separate the males and females, it was decided to put males and females together during blood-feeding. Subsequent selection was done by taking out the blood-fed females, leaving the non-blood-fed females and males out.
- In the results, the authors state that the "ratio of pipiens to molestus biotype is 12/63", and again below. Is this the correct ratio?
The ratios are correct, at the constant temperature condition there were 76 mosquitoes infected in total, of these 63 belonging to the biotype pipiens, 12 to biotype molestus and 1 hybrid species.
- Figure three requires an uninfected control for comparison to appreciate the staining pattern.
Implemented. The following was added to the caption: JEV was isolated from the saliva of a Culex pipiens mosquito collected at 14 dpi from the constant 25°C temperature condition (left panel). Saliva of a non-blood fed mosquito was included as a negative control (right panel).
- At the end of the 1st paragraph of the Discussion, suggest "feeding" rather than "drinking".
Implemented.
- The authors state that they hypothesize that these clear differences between field-collected and colonized mosquitoes are due to microbiome changes and/or a lower genetic diversity. What exactly is this hypothesis based on? They then go on to state that "This has been proven for other mosquito-arbovirus combinations [27-30]." The references to the influence of the microbiome are reviews and a book chapter. These base these assumptions on experiments crudely treating mosquitoes with antibiotics. The authors should refer to original research that demonstrates that colonisation influences that mosquito microbiome (demonstrated rather than implied) to a degree that changes vector competence.
We thank the reviewer for this valid criticism. The following was added to the discussion (line 314-319): These differences between field-collected and colonized mosquitoes have already been reported for other mosquito species. The bacterial diversity in field-collected Aedes albopictus mosquitoes was found to be much higher compared to colonized lines [29]. Additionally, it was demonstrated that colonized Aedes aegypti mosquitoes are less genetically variable than their field counterparts [30].
Based on these original research articles:
- Tuanudom R, Yurayart N, Rodkhum C, Tiawsirisup S. Diversity of midgut microbiota in laboratory-colonized and field-collected Aedes albopictus (Diptera: Culicidae): A preliminary study. Heliyon. 2021;7(10):e08259. Published 2021 Oct 27. doi:10.1016/j.heliyon.2021.e08259
- Gloria-Soria A, Soghigian J, Kellner D, Powell JR. Genetic diversity of laboratory strains and implications for research: The case of Aedes aegypti. PLoS Negl Trop Dis. 2019;13(12):e0007930. Published 2019 Dec 9. doi:10.1371/journal.pntd.0007930
- In the following paragraph there are numerous references to midgut escape barrier. What evidence do the authors have for this influencing virus replication in their model? Reduced temperature may have influenced virus replication although the implication of Figure 2 suggests that this is not the case. The lower temperature regime may have delayed virus egress rather than prevented it, but only one time point was used. Overall the findings of this study seem similar to Folly et al., 2021.
We thank the reviewer for their justified comment. Indeed, reduced replication in the midgut was not demonstrated in our study, as the infection rate did not differ significantly between the two conditions. Therefore, the following was added to the discussion (line 345-349): As a last point, all vector competence parameters in this study were determined after an incubation time of 14 days. It would be interesting to evaluate whether these would remain similar after a longer incubation period, e.g. 21 days, and study whether the dissemination is only delayed in the gradient condition rather than inhibited.
The study of Folly et al. can indeed be compared with ours since they also show no transmission at the lower temperature condition, though in their study there is a significant difference between infection at both conditions, which is not the case in ours.
- In the conclusions, the data "suggests" rather than "makes" that under current conditions. Also the authors should consider extreme weather events where temperatures are elevated for extended periods. This appears to be a consequence of climate change. Finally, another flavivirus, has been transmitted successfully in Belgium, can the authors comment?
Implemented. The following was added to the conclusions (line 421-422): or extreme weather events involving high temperatures for extended periods of time.
No other vector competence studies for any flavivirus were conducted in Belgium up to now. Regarding the detection of other flaviviruses in Belgium, there are 3 confirmed cases of autochthonous Tick Borne encephalitis virus that all occurred in Belgium in 2020 (ref.: Stoefs A, Heyndrickx L, De Winter J, et al. Autochthonous Cases of Tick-Borne Encephalitis, Belgium, 2020. Emerg Infect Dis. 2021;27(8):2179-2182. Doi:10.3201/eid2708.211175) and Usutu virus was epizootic in Belgium in 2017 and 2018, with mainly detection in blackbirds (Benzarti E, Sarlet M, Franssen M, et al. Usutu Virus Epizootic in Belgium in 2017 and 2018: Evidence of Virus Endemization and Ongoing Introduction Events. Vector Borne Zoonotic Dis. 2020;20(1):43-50. Doi:10.1089/vbz.2019.2469). Usutu infections in humans have not been detected in Belgium. As Usutu virus is transmitted by mosquitoes, vector competence studies with indigenous mosquitoes would expose the potential vectors. Additionally, large-scale screening of mosquitoes in Belgium, would reveal the confirmed vectors and provide valuable information about the prevalence and distribution of Usutu virus in Belgium.